# A Framework for Generating High Spatiotemporal Resolution Land Surface Temperature in Heterogeneous Areas

Xinming Zhu [1,2], Xiaoning Song [1,2,*], Pei Leng [3,4], Xiaotao Li [5], Liang Gao [1,2], Da Guo [1,2] and Shuohao Cai [1,2]

1 College of Resources and Environment, University of Chinese Academy of Sciences, Beijing 101408, China; zhuxinming19@mails.ucas.ac.cn (X.Z.); gaoliang17@mails.ucas.ac.cn (L.G.); guoda181@mails.ucas.ac.cn (D.G.); caishuohao19@mails.ucas.ac.cn (S.C.)
2 Yanshan Earth Critical Zone and Surface Fluxes Research Station, University of Chinese Academy of Sciences, Beijing 101408, China
3 Key Laboratory of Agricultural Remote Sensing, Ministry of Agriculture and Rural Affairs, Beijing 100081, China; lengpei@caas.cn
4 Institute of Agricultural Resources and Regional Planning, Chinese Academy of Agricultural Sciences, Beijing 100081, China
5 China Institute of Water Resources and Hydropower Research, Beijing 100038, China; lixt@iwhr.com
* Correspondence: songxn@ucas.ac.cn; Tel.: +86-186-1158-9622

**Abstract:** Land surface temperature (LST) is a crucial biophysical parameter related closely to the land–atmosphere interface. Satellite thermal infrared measurement provides an effective method to derive LST on regional and global scales, but it is very hard to acquire simultaneously high spatiotemporal resolution LST due to its limitation in the sensor design. Recently, many LST downscaling and spatiotemporal image fusion methods have been widely proposed to solve this problem. However, most methods ignored the spatial heterogeneity of LST distribution, and there are inconsistent image textures and LST values over heterogeneous regions. Thus, this study aims to propose one framework to derive high spatiotemporal resolution LSTs in heterogeneous areas by considering the optimal selection of LST predictors, the downscaling of MODIS LST, and the spatiotemporal fusion of Landsat 8 LST. A total of eight periods of MODIS and Landsat 8 data were used to predict the 100-m resolution LST at prediction time $t_P$ in Zhangye and Beijing of China. Further, the predicted LST at $t_P$ was quantitatively contrasted with the LSTs predicted by the regression-then-fusion strategy, STARFM-based fusion, and random forest-based regression, and was validated with the actual Landsat 8 LST product at $t_P$. Results indicated that the proposed framework performed better in characterizing LST texture than the referenced three methods, and the root mean square error (RMSE) varied from 0.85 K to 2.29 K, and relative RMSE varied from 0.18 K to 0.69 K, where the correlation coefficients were all greater than 0.84. Furthermore, the distribution error analysis indicated the proposed new framework generated the most area proportion at 0~1 K in some heterogeneous regions, especially in artificial impermeable surfaces and bare lands. This means that this framework can provide a set of LST dataset with reasonable accuracy and a high spatiotemporal resolution over heterogeneous areas.

**Keywords:** land surface temperature; spatiotemporal resolution; heterogeneity; random forest; image fusion





## 1. Introduction

Land surface temperature (LST) is a crucial terrestrial geophysical variable that affects the heat transformation process between the land surface and the atmospheric boundary layer [1]. Its spatiotemporal dynamics play an important role in impacting the surface energy balance, soil moisture content, evapotranspiration, and surface thermal environment [2,3]. As such, continuous spatiotemporal estimation of LST is essential for related fields of terrestrial surface process on a regional or global scale, such as soil moisture content monitoring, vegetation evaporation estimation, water and heat flux measurement, and urban heat island (UHI) monitoring [4].

Thermal infrared (TIR) technology based on the polar-orbiting satellite observations provides an alternative method for obtaining LST data at different temporal and spatial scales [5]. The famous TIR sensors mainly include the Landsat 5 Thematic Mapper (TM), the Landsat 7 Enhanced Thematic Mapper Plus (ETM+), the Landsat 8 Thermal Infrared Sensor (TIRS), the Advanced Spaceborne Thermal Emission and Reflection Radiometer (ASTER), the Visible Infrared Imaging Radiometer Suite (VIIRS), the Advanced Very High Resolution Radiometer (AVHRR), and the Moderate Resolution Imaging Spectroradiometer (MODIS) [6]. However, owing to the trade-off between spatial and temporal resolutions in TIR sensors, these sensors either provide high-resolution LST data with a longer revisiting period or low-resolution LST data with a quick revisiting period [7]. For instance, TIR images of Landsat series sensors can provide some LST images with a spatial resolution of 60~120 m, but the revisiting cycle is 16 days; in contrast, MODIS LST products have a 1-km resolution, whereas the revisiting cycle is four times per day. Obviously, it is very difficult to obtain LST with high spatial and temporal resolutions at the same time using a single TIR sensor [8], which greatly confines potential research in the global environment and ecology fields. Especially in the research field of agricultural yield estimation, at the critical crop growth stage, the continuous and rapid monitoring of agricultural drought, evapotranspiration, and soil heat change urgently requires the LST data at both high spatial resolutions and short revisiting periods [9–12].

Acquiring the LST with high spatiotemporal resolutions is an urgent task at present. In recent years, two major categories of approaches have been widely used to cope with this problem, including the LST downscaling and spatiotemporal image fusion [13]. The LST downscaling is a simple method to generate high spatiotemporal resolutions LST. This method is mainly used to sharpen the spatial resolution of low-resolution daily LST such as MODIS LST. It first aggregates high-resolution LST predictors (e.g., normalized difference vegetation index, NDVI; surface albedo, $\alpha$; normalize difference build-up index, NDBI; normalized difference water body index, NDWI; land surface emissivity, LSE; land-use/land-cover, LULC; digital elevation model, DEM) to the spatial resolution of MODIS LST and then uses different statistical regression models to establish a linking model between MODIS LST and upscaled LST predictors at coarse resolution. Finally, this constructed regression model is then applied to the initial LST predictors for generating high-resolution LST by assuming that the regression relationship between MODIS LST and its predictors is scale-invariant at various resolution scales [14]. Representative methods include the Thermal sHARPening (TsHARP) algorithm [15], the high resolution urban thermal sharpener (HUTS) algorithm [16], the non-linear DisTrad (NL-DisTrad) algorithm [17], geographical weighted regression-based (GWR-based) algorithm [6], and the Random Forest-based (RF-based) algorithm [18]. In contrast, the spatiotemporal image fusion is a more reliable method than the LST downscaling since it can inherit the spatial image textures from high-resolution LST, including Landsat LST and the variation information determined by low-resolution LST images such as MODIS LST from the start time to the end time [19–22]. However, different from the traditional image fusion methods, such as the wavelet transform, principal component analysis (PCA), and intensity-hue-saturation transformation (IHS), this method requires at least one pair of Landsat LST and MODIS LST images at a base time $t_b$ (i.e., the time with the high-resolution LST) and a series of MODIS LST image at the prediction time $t_p$ (i.e., the time without the high-resolution LST) as data input [23]. Since Gao et al. [24] proposed a spatial and temporal adaptive reflectance fusion model (STARFM) for detecting the land surface reflectance changes of various land-cover types, many spatiotemporal image fusion models have been widely proposed to fuse high-resolution LST time series, such as the spatial-temporal adaptive data fusion algorithm for temperature mapping (SADFAT) model [19], spatiotemporal integrated temperature fusion model (STITFM) [21], wavelet artificial intelligence fusion approach (WAIFA) [25], and deep learning-based spatiotemporal temperature fusion network (STTFN) [11].

However, the LST spatial downscaling can only sharpen MODIS LST when there are high-resolution LST predictors and has poorer performance when the ratio of the

fine/coarse resolution is too large [26,27]. The spatiotemporal data fusion method could produce long time-series LSTs but is limited by the resolution of high-resolution LST [28]. Thus, making full use of the advantages of LST downscaling and spatiotemporal image fusion to generate LST with high spatiotemporal resolutions has attracted much attention in recent years. At present, many scholars have developed some hybrid strategies [23,28]. For instance, Bai et al. [28] used the extreme learning machine algorithm to first downscale the Landsat ETM+ TIR to 30-m and then used the SADFAT to fuse MODIS LST time series and the downscaled LST for generating 30-m resolution LST time series. Xia et al. [23] proposed the regression-then-fusion (RF) strategy, applied the RF algorithm to downscale the Landsat 8 LST data to the 30-m resolution, and then used the downscaled LST as input for the STARFM to sharpen MODIS LSTs. These methods blend the advantages of regression and data fusion and present better performances than the first two kinds of methods. In spite of this, some key issues still need to be fully considered for the hybrid strategy. On the one hand, due to the large difference in resolution between MODIS LST and Landsat LST, if the MODIS LST is directly re-sampled to fine pixels within the 100-m range for subsequent spatiotemporal data fusion, a large amount of LST change information will not be captured. On the other hand, since the previous hybrid strategies usually employed the STARFM and SADFAT algorithms that do not fully consider the LST pattern heterogeneity of complex landscapes to produce the high-resolution LST time-series, the LSTs generated over the heterogeneous surface usually present out poor accuracy. As a result, it is very necessary to develop a more effective method to gain accurate high spatiotemporal resolution LSTs in non-uniform regions.

According to the above statement, the main objective of this study is to develop a new framework for producing an LST dataset with reasonable accuracy and a high spatiotemporal resolution. Similar to the RF strategy, the proposed new framework also will blend the regression-based LST downscaling process and the spatiotemporal image fusion technology. However, different from the previous RF strategy, this framework will take into full account the optimal selection of LST predictors, the downscaling of MODIS LST, and the spatiotemporal fusion of Landsat 8 LST in complex landscapes for maintaining the accuracy and detailed texture of LST image. As a whole, there are the following two advantages: (1) it can greatly improve the LST prediction accuracy using the downscaling of MODIS LST to better capture the change in MODIS LST from basic time $t_b$ to target time $t_p$; and (2) it can acquire fine spatial texture of LST in the heterogeneous landscape using the flexible spatiotemporal data fusion (FSDAF) algorithm that considers the landscape heterogeneity of land surface fully.

The rest of this article is organized as follows. Section 2 introduces our study area and data set. Section 3 describes the establishment process of this framework. Section 4 shows the results, and Sections 5 and 6 give the discussions and conclusions, respectively.

## 2. Study Area and Data Collection

### 2.1. Study Area

A total of two areas with complex landscapes in China were selected as cases in our study in order to make our framework more representative. Figure 1 presents the geolocations of the study areas with two LULC maps generated from the global land cover (GLC) dataset.



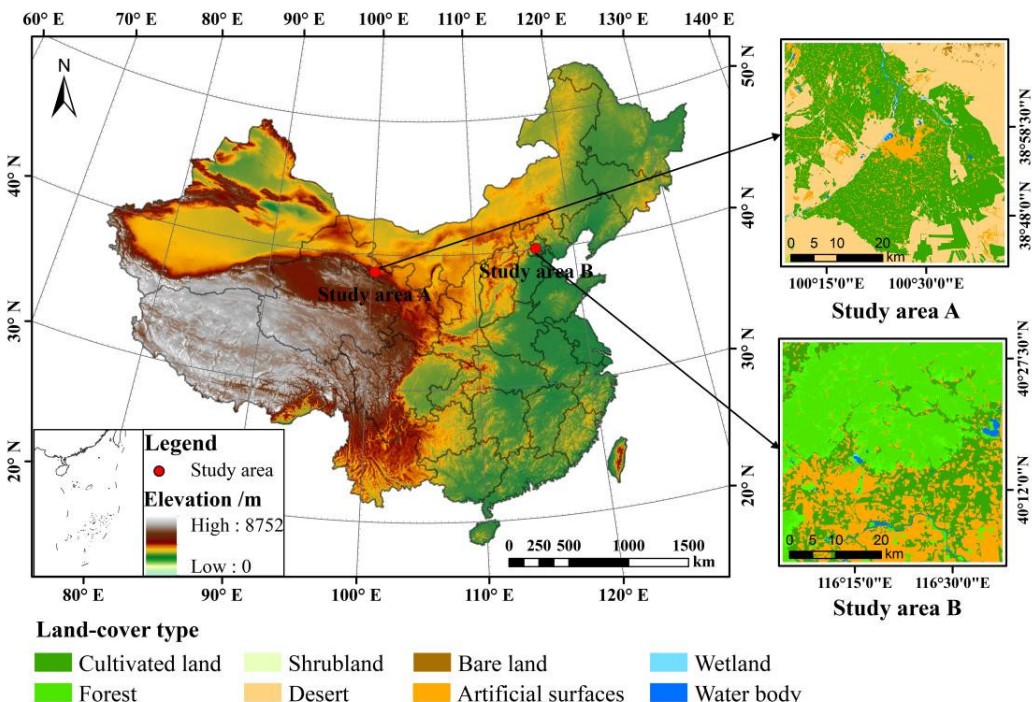

**Figure 1.** Geographical locations and land-cover maps of study areas (The land cover maps in two study areas are acquired from the 30-m resolution global land cover dataset).

Study area A is located in Zhangye of Gansu Province in China, which belongs to the middle reach of the Heihe River Basin. The coordinate range is between latitudes 38°42′ N and 39°08′ N and longitudes 100°08′ E and 100°41′ E. This region possesses a temperate continental arid climate type, the annual average temperature is approximately 7 °C, and the annual average precipitation is less than 200 mm. The topography is high in the northeast part and southwest part and low in the central region. From the perspective of land-cover type, this region is mainly characterized by cultivated lands, deserts, bare lands, and artificial surfaces. In recent years, with the continuous implementation of the Heihe Watershed Allied Telemetry Experimental Research (Hi-WATER), this region is frequently selected as the case to implement the scientific experiments on satellite and ground observations [22,29].

Study area B is located in Beijing of China, which belongs to the transitional zone between the North China Plain and the Inner Mongolian Plateau. The coordinate range is between latitudes 40°03′ N and 40°30′ N and longitudes 116°03′ E and 116°37′ E. This region has a warm temperate continental climate type, the annual average temperature is approximately 10~12 °C, and the annual average precipitation is approximately 483.9 mm. The topography is high in the north part and low in the south part. Regarding the land-cover type, this region is mainly characterized by artificial surfaces, cultivated lands, forest lands, and water bodies. Among them, urban artificial surfaces and forest lands are two kinds of main land-cover types, and they are widely distributed from the north part of the study area and the south part of the study area, respectively. Because the landscape type of this region is complex, this study area is of significant meaning for our experiment.

### 2.2. Data Collection and Image Processing

Since the Earth observation time of Terra and Landsat 8 satellites are similar, this paper used eight periods of MODIS and Landsat 8 remote sensing images collected in the two study areas to realize the construction of this framework. Study area A used four periods of satellite images collected on 5 July 2013, 21 July 2013, 24 July 2014, and 9 August 2014 (Image ID: A1, A2, A3, A4). Study area B used four periods of satellite images acquired on 4 September 2014, 6 October 2014, 12 September 2017, and 28 September 2017 (Image ID:

B1, B2, B3, B4). All selected satellite images were collected under clear sky conditions, and Table 1 shows the information of the data set used.

**Table 1.** The used data set and their main information.

| Satellite | Data Collection | Factors Provided | Spatial Resolution | Temporal Resolution |
|---|---|---|---|---|
| Terra/MODIS | MOD11A1 | LST, LSE | 1-km | 1 day |
| | MOD09GQ | NDVI, PV | 250-m | 1 day |
| | MOD09GA | NDVI, PV, SAVI, NDMI, NDBI, BSI, IEI, MNDWI | 500-m | 1 day |
| | MCD12Q1 | LULC | 500-m | 1 year |
| Landsat 8 | RTU LST product | LST | 100-m | 16 days |
| ASTER | ASTER GDEM | longitude, latitude, elevation, slope, aspect | 30-m | |

(1) MODIS product

The MODIS sensor provided plentiful satellite products for understanding the surface change at the global scale and has a moderate spatial resolution with daily continuous global coverage. In this study, the MOD11A1 (collection 6) was used to provide the LST data, and the MOD09GA and MCD12Q1 were used to extract the candidate predictors. The MOD09GQ was used to calculate the final predictors for performing the MODIS LST downscaling. All selected MODIS products were downloaded from the Next Generation Earth Science Discovery Tool (https://ladsweb.modaps.eosdis.nasa.gov/search/, accessed on 28 September 2021).

The MOD11A1 includes the pixel-by-pixel LST and LSE with a 1-km resolution in a sequence of swath-based to grid-based global products, whose LST is derived from the channels 31 and 32 of MODIS using the generalized split-window algorithm [30]. Previous studies have indicated that the estimated LST has good accuracy with less than 1.3 K for most homogeneous surfaces, which has been widely used in LST change analysis [31]. The MOD09GA provides seven bands with 500-m resolution in the Sinusoidal projection. The MCD12Q1 provides seventeen kinds of land-cover information with a 500-m resolution each year. The MOD09GQ provides two bands with 250-m resolution in the Sinusoidal projection.

(2) Landsat 8 product

Landsat 8 data provides eight 30-m resolution visible and infrared bands and two 100-m resolution thermal infrared bands for the Earth's monitoring, but there is a relatively long revisiting period of 16 days. In this study, the Ready-To-Use (RTU) Landsat 8 LST product provided by the Chinese Academy of Sciences was used for helping the spatiotemporal fusion of LST and was further used as an actual reference for the evaluation of the predicted LSTs. The RTU product can be acquired from the DATABANK Remote Sensing Data Engine (http://databank.casearth.cn, accessed on 28 September 2021).

The RTU LST product is produced with the generalized single-channel (GSW) algorithm and covers China and central Asia. It provides the LST estimate after 2000 [32]. Previous studies have indicated the comparison between the RTU LST product and in-situ LST measurement in three regions (Xuanwu Lake, Zoucheng, Huairou of China) shows good accuracy, with an average RMSE of 0.83 K [33]. Thus, it can be ensured the RTU LST product can reveal the actual distribution condition of LST to some extent.

(3) DEM image

Because the LST distribution also depends on the geographical location and topographic factors [34], the DEM derived from the ASTER GDEM was collected in this paper to yield five LST predictors: elevation, slope, aspect, longitude, and latitude. In this paper, the GDEM data were collected from the Center for Earth Observation at

Yale University (https://yceo.yale.edu/aster-gdem-global-elevation-data/, accessed on 28 September 2021).

The GDEM data were mainly generated from the ASTER sensor onboard the Terra and has been applied in topography studies from 83° S latitude to 83° N latitude. It has a resolution of 30 m with an absolute vertical error of less than 20 m [35].

(4)    Image processing

Using the MODIS re-projection tool (MRT), all MODIS products in the HDF-EOS format were re-projected to the Universal Transverse Mercator (UTM) WGS-1984 projection and re-sampled to 1-km, 500-m, 500-m, and 250-m resolutions. In addition, for better matching the geographic location between MODIS product and Landsat 8 product, the collected MODIS products were all geo-referenced to the locations of the RTU LST products by selecting control points such as road and river intersections using the image-to-image module of ENVI 5.3 software.

## 3. Methodology

### 3.1. Overview

The proposed framework mainly consists of three steps, including (1) the optimal selection of LST predictors; (2) the downscaling of MODIS LST product; and (3) the spatiotemporal image fusion of Landsat 8 LST at $t_p$. For simplicity, we refer to the new framework as the three-step method for short. The detailed implementation is illustrated in Figure 2.

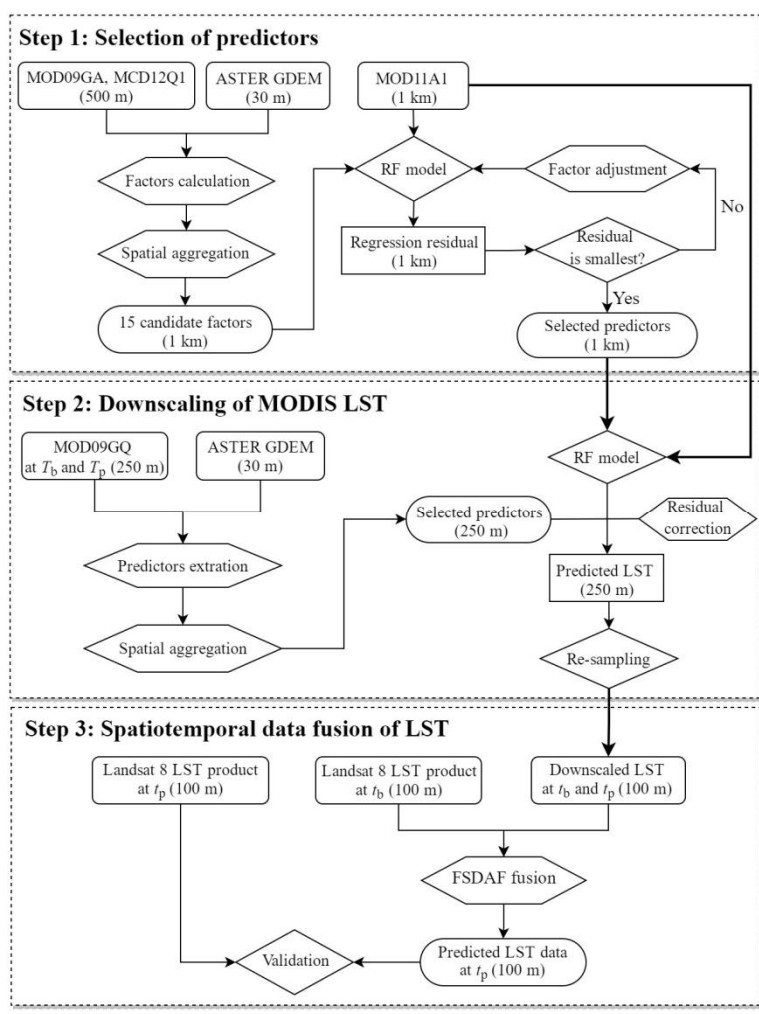

**Figure 2.** Implementation flowchart of the new framework.

To be specific, in step 1, LST predictors used for downscaling MODIS LSTs are determined according to the variable importance in the RF algorithm. In step 2, to better capture the change information of MODIS LST from $t_b$ to $t_p$, the MODIS LSTs at $t_b$ and $t_p$ are sharpened to 250-m by using the RF-based LST downscaling. In step 3, after the 100-m resolution RTU Landsat 8 LST product at $t_b$ is obtained, the downscaled 250-m MODIS LSTs at $t_b$ and $t_p$ are together imported into the FSDAF algorithm to generate the 100-m Landsat 8-like LST at $t_p$. Finally, the actual RTU LST at $t_p$ is used to evaluate the predicted LST at $t_p$ objectively. For each study area, we used the Terra/MODIS LST products and Landsat 8 remotely sensed images from the same year with different basic and prediction times to achieve this framework. One input date pair is regarded as the base data at $t_b$, while the other is used as the prediction data at $t_p$. Using the proposed three-step method, we can obtain the Landsat 8-like LST at $t_p$ given one LST date pair at $t_b$ and one downscaled MODIS LST at $t_p$. The specific implementation of this framework is expounded in Section 3.2.

### 3.2. Construction of the Framework

Step 1: Selection of LST predictors

The choice of LST predictors plays an essential role in implementing the spatial downscaling of MODIS LST. Since the LST in heterogeneous areas is strongly affected by various biophysical factors, a large number of predictors associated with LST have been widely adopted to characterize LST change in previous studies [14]. However, selecting a number of LST predictors to reveal LST patterns can lead to data redundancy and multicollinearity among variables. In addition, this process will waste lots of time to establish the LST downscaling model. Thus, in order to provide a relatively robust LST downscaling model, this step aims to determine the optimal LST predictors from a number of auxiliary parameters by reducing the redundancy of relevant factors.

By referring to a large number of studies [8,18,36–41], our study selected a total of fifteen parameters significantly correlated with LST as candidate factors to determine reliable LST predictors. Eight remotely sensed indexes impacting the LST were extracted from the MOD09GA (i.e., NDVI; percent vegetation, PV; the normalized difference moisture index, NDMI; bare soil index, BSI; the soil adjusted vegetation index, SAVI; NDBI; integrated ecological index, IEI; and MNDWI). Five terrain elements that are highly correlated with LST differentiation were acquired from the GDEM (i.e., elevation, slope, aspect, longitude, and latitude). The land-cover type that effectively reveals the change in LSTs was acquired from the MCD12Q1 product. In view of the importance of LSE in the process of LST retrieval, LSE was also calculated using the NDVI threshold method proposed by Sobrino et al. [42]. Partial factors were calculated as follows:

$$\text{NDVI} = \frac{\text{NIR} - \text{RED}}{\text{NIR} + \text{RED}} \tag{1}$$

$$\text{SAVI} = \frac{(\text{NIR} - \text{RED})(1 + 0.5)}{(\text{NIR} + \text{RED} + 0.5)} \tag{2}$$

$$\text{PV} = \left[ \frac{\text{NDVI} - \text{NDVI}_{\min}}{\text{NDVI}_{\max} - \text{NDVI}_{\min}} \right]^2 \tag{3}$$

$$\text{NDMI} = \frac{\text{NIR} - \text{SWIR}_2}{\text{NIR} + \text{SWIR}_2} \tag{4}$$

$$\text{BSI} = \frac{(\text{SWIR}_1 + \text{RED}) - (\text{BLUE} + \text{NIR})}{(\text{SWIR}_1 + \text{RED}) + (\text{BLUE} + \text{NIR})} \tag{5}$$

$$\text{NDBI} = \frac{\text{SWIR}_1 - \text{NIR}}{\text{SWIR}_1 - \text{NIR}} \tag{6}$$

$$\text{MNDWI} = \frac{\text{GREEN} - \text{SWIR}_1}{\text{GREEN} + \text{SWIR}_1} \tag{7}$$

$$\begin{aligned}\text{IEI} \quad &= 1 - PC_1\{[f(\text{SAVI}, \text{NDMI}, \text{BSI}, \text{NDBI})]\} \\ &= 1 - \sum_{i=1}^{m} a_i PC_i\end{aligned} \tag{8}$$

$$\varepsilon = \begin{cases} 0.973\,, \text{NDVI} < 0.05 \\ 0.99\,, \text{NDVI} > 0.7 \\ 0.004\text{PV} + 0.986,\ 0.05 \leq \text{NDVI} \leq 0.7 \end{cases} \tag{9}$$

where NIR is the near-infrared band reflectance; RED is the red band reflectance; $\text{SWIR}_1$ is the shortwave infrared band reflectance in 1.560~1.660 μm; BLUE is the blue band reflectance; $\text{SWIR}_2$ is the shortwave infrared band reflectance in 2.100~2.300 μm; and GREEN is the green band reflectance. IEI is the integrated ecological index proposed by Zhu et al. [43], which is calculated using the PCA with SAVI, NDMI, BSI, and NDBI indexes, and has been normalized to the range of 0~1. $PC_1$ is the first principal component of PCA; $a_i$ is the variance contribution weight of the principal component; $PC_i$ is the first principal component for each ecological factor; and $m$ is the number of surface ecological factors.

Since the multicollinearity has little effect on the predictive ability of the RF regression model, this article used the variable importance score of each candidate LST predictor to select the optimal LST predictors [44]: (1) the resolutions of fifteen candidate factors were resampled to 1 km using the pixel aggregation tool of ENVI software; (2) corresponding pixels from the 1-km resolution factors and the MODIS LST were selected; (3) after the outlier removal, RF regression was used to implement the non-linear fitting one-by-one by removing or replacing certain factor; and (4) after the variable importance of fifteen factors were all calculated, the factors with a high Gini index were chose to implement the non-linear regression until the residual up to the minimum; (5) via the incessant tests, the optimal LST predictors were determined based on the fitting residual of RF regression model. In this paper, the selected LST predictors were fixed as the PV, elevation, slope, longitude, and latitude in the end.

Step 2: Downscaling of MODIS LST

Since previous fusion methods usually re-sample the MODIS LST data from 1-km resolution to the resolution within 100-m for the subsequent spatiotemporal data fusion, this process will lose detailed spatiotemporal change information of LST. Fortunately, the Terra/MODIS sensor can provide two kinds of resolutions PV images (i.e., 250-m and 500-m) every day to assist the downscaling of MODIS LST. Meanwhile, these PV images have the same instantaneous observation time as MODIS LST images. In our study, for better capturing the spatial texture information of LST change and enhancing the performance of LSTs fusion procedures in the following process, we chose the 250-m resolution PV image derived from Terra/MODIS to sharpen MODIS LSTs from 1-km to 250-m resolutions. This is because that the introduction of a 250-m resolution meets the requirement that the resolution difference is less than 3~5 times in the spatially non-uniform surfaces [26] and offers more abundant spatiotemporal change information of LST in the process of LST downscaling.

In previous MODIS LST downscaling studies, GLR, MLR, and GWR regression models have been extensively applied to estimate model coefficients and then to predict the high-resolution LSTs with calibrated parameters and LST predictors [14]. However, under the assumption that the local atmospheric condition is homogeneous, the physic mechanism of a linear regression model is very hard to apply in complex regions due to the changes in land-cover types. Since the RF algorithm has remarkable performance in automatically settling the non-linear relationship between the LST and its predictors by constructing and averaging a large number of randomized and de-correlated decision trees [18], in this study, we used the RF-based non-linear regression model to implement the downscaling of

MODIS LST. With the selected LST predictors in step 1, the detailed process of the MODIS LST downscaling is displayed in Figure 2 and is summarized as follows:

(1) The 250-m resolution MOD09GQ product and 30-m resolution GDEM were used to calculate the selected LST predictors and then were aggregated to 1 km and 250 m, respectively. LST predictors with a resolution of 1 km belong to the MOD11A1 pixel level, and LST predictors with a resolution of 250 m belong to the MOD09GQ pixel level.

(2) The RF regression model was used to construct the relationship between MODIS LST and five predictors at the resolution of 1 km, which can be expressed as follows:

$$LST_{1km} = f(PV_{1km}, elevation_{1km}, slope_{1km}, longitude_{1km}, latitude_{1km}) + \varepsilon_{1km} \quad (10)$$

where $LST_{1km}$ denotes the MODIS LST and is fitted by the RF regression with $f$ as a non-linear function; $f$ denotes the function between LST and its predictors; $PV_{1km}$ is the aggregated PV image using the MOD09GQ with the resolution of 1 km; $elevation_{1km}$, $slope_{1km}$, $longitude_{1km}$ and $latitude_{1km}$ are all the aggregated LST predictors derived from the GDEM with the resolution of 1 km; and $\varepsilon_{1km}$ is the residual of RF regression at a spatial resolution of 1 km.

(3) By assuming that regression residuals are uniformly distributed in space, the ordinary kriging interpolation were used to interpolate the residual with a 1-km resolution to 250 m.

(4) By assuming that the relationship between LST and its predictors within 1-km resolution is scale-invariant for 250-m resolution, the MODIS LST was sharpened at $t_b$ and $t_p$ to 250 m based on the linking model at 1-km resolution and combined with the residual and predictors at 250-m resolution:

$$LST_{250m} = f\left(PV_{250m}, elevation_{250m}, slope_{250m}, longitude_{250m}, latitude_{250m)}\right) + \varepsilon_{250m} \quad (11)$$

where $LST_{250m}$ is the downscaled MODIS LST with a resolution of 250 m; $PV_{250m}$ is the aggregated PV from the MOD09GQ with a resolution of 250 m; $elevation_{250m}$, $slope_{250m}$, $longitude_{250m}$, and $latitude_{250m}$ are all the aggregated terrain factors derived from the GDEM with a resolution of 250 m; $\varepsilon_{250m}$ is the regression residual with a resolution of 250 m.

Step 3: Spatiotemporal image fusion of LST.

After the MODIS LST products at $t_b$ and $t_p$ were downscaled to a 250-m resolution, and the FSDAF algorithm was used to predict the Landsat 8-like LST product at $t_p$ in combination with the RTU LST product at $t_b$. Initially, this algorithm was mainly used to fuse land surface reflectance with the high spatiotemporal resolutions in heterogeneous areas using daily low-resolution MODIS surface reflectance and high spatial resolution reflectance. Later on, many studies also used this algorithm to estimate the other surface physical parameters (e.g., NDVI and LST). This algorithm makes full use of the image texture details from the neighboring pixels and effectively considers the abrupt land-cover type changes in the heterogeneous regions. It requires the same input data as two widely used spatiotemporal fusion methods, including STAFRM [24] and STITFM algorithms [21], whereas its prediction performance is more accurate than the STAFRM and STITFM by uniting the advantages of spectral unmixing analysis and a thin plate spline (TPS) interpolator [45].

The LST spatiotemporal fusion based on the FSDAF is mainly implemented by using six steps as follows [45]: (1) classifying the RTU LST product at $t_b$ into five main levels based on the SVM classifier; (2) estimating the temporal change information of each LST level in the downscaled MODIS LST image from $t_b$ to $t_p$; (3) predicting the Landsat 8-like LST at $t_p$ using the class-level temporal change information, and calculating the residuals at each downscaled MODIS LST pixels; (4) using the TPS interpolation to predict the Landsat 8-like LST at $t_p$ from the downscaled MODIS LST at $t_p$; (5) distributing the residuals at each

pixel of the downscaled MODIS LST using the TPS prediction; and finally (6) acquiring the prediction image of Landsat 8-like LST using the information in neighboring pixels.

Specifically, the equation used to fuse the Landsat 8-like LST at $t_p$ through the FSDAF algorithm is given as follows:

$$\hat{F}_2(x_{ij}, y_{ij}, \text{LST}) = F_1(x_{ij}, y_{ij}, \text{LST}) + \sum_{k-1}^{n} w_k \times \Delta F(x_k, y_k, \text{LST}) \tag{12}$$

where $\hat{F}_2(x_{ij}, y_{ij}, \text{LST})$ is the final LST prediction value of the target pixel $(x_{ij}, y_{ij})$ at $t_p$; $F_1(x_{ij}, y_{ij}, \text{LST})$ is the LST value of the $j$-th fine pixel (i.e., Landsat 8 LST) within the coarse pixel at location $(x_i, y_i)$ observed at $t_b$; $n$ is the number of similar pixels for $(x_{ij}, y_{ij})$ in a sliding window; $w_k$ is the weight for the $k$-th similar pixel; and $\Delta F(x_k, y_k, \text{LST})$ is the LST prediction of the total change in a fine pixel between $t_b$ and $t_p$.

$w_k$ is an important parameter for the spatiotemporal data fusion and is determined by the spatial distance between similar pixels and the target pixel with:

$$w_k = (1/D_k) / \sum_{k=1}^{n} (1/D_k) \tag{13}$$

$$D_k = 1 + \sqrt{(x_k - x_{ij})^2 + (y_k - y_{ij})^2} / (w/2) \tag{14}$$

where $w_k$ is the weight; $D_k$ is a relative distance ranging from 1 to $\sqrt{2}$; $(x_k, y_k)$ and $(x_{ij}, y_{ij})$ denote the target pixel and similar pixels in a sliding window, respectively; and $w$ is the size of the neighborhood, which is determined by the homogeneity of the study area and the size of coarse pixels. The schematic diagram of similar pixels in a moving window is displayed in Figure 3.

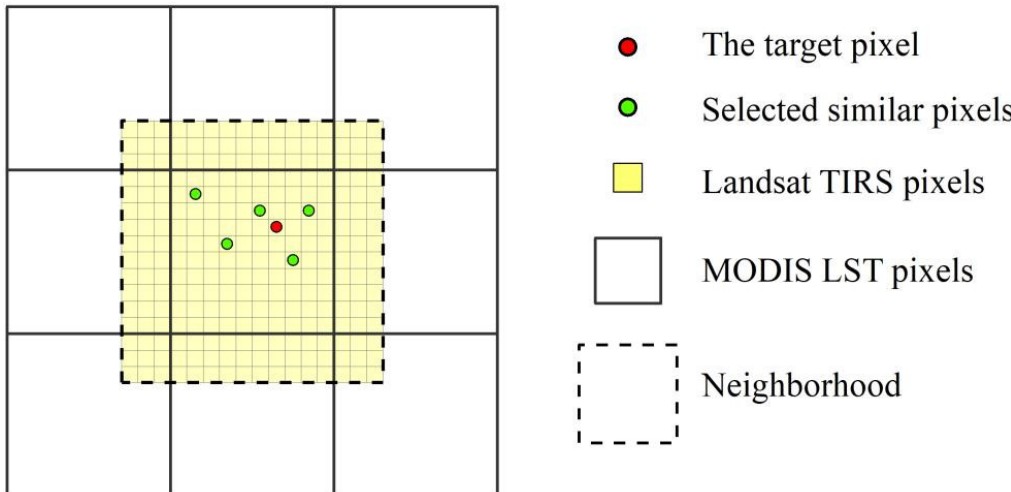

**Figure 3.** Schematic diagram of similar pixels selected in a neighborhood range of the target pixel.

$\Delta F(x_{ij}, y_{ij}, \text{LST})$ depends on the changes in MODIS LSTs from $t_b$ to $t_p$, which can be estimated using the distributed residual $r$ and temporal change information $\Delta F$:

$$\Delta F(x_{ij}, y_{ij}, \text{LST}) = r(x_{ij}, y_{ij}, \text{LST}) + \Delta F(L, \text{LST}) \tag{15}$$

where $r(x_{ij}, y_{ij}, \text{LST})$ is the LST residual distributed to the $j$-th fine pixel; $\Delta F(\text{LST})$ is the changes in LSTs of different LST levels ($L$) at fine resolution from $t_b$ to $t_p$.

Among them, $r(x_{ij}, y_{ij}, \text{LST})$ can be further described as follows:

$$r(x_{ij}, y_{ij}, \text{LST}) = m \times R(x_i, y_i, \text{LST}) \times W(x_{ij}, y_{ij}, \text{LST}) \tag{16}$$

where $m$ is the number of fine pixels (also named as subpixels) within one coarse pixel; $R(x_i, y_i, \text{LST})$ is a residual term between the true values and temporal prediction of fine pixels; $W(x_{ij}, y_{ij}, \text{LST})$ is the weight of residual distribution.

For a more detailed description of the FSDAF algorithm, we can refer to Zhu et al. [45].

### 3.3. Comparison with Other Methods

For highlighting the proposed framework, three LST prediction approaches were additionally applied in this study to serve as contrasts: (1) the RF strategy; (2) the STARFM-based fusion; and (3) the RF-based LST downscaling. These three methods, respectively, represent the hybrid strategy, spatiotemporal data fusion method, and LST downscaling method. They all have obvious advantages within their respective fields.

Different from the previous RF strategy, in our study, the RF strategy used the RF regression model to downscale the MODIS LSTs at $t_b$ and $t_p$ to 250-m resolution based on the NDVI, then adopted the STARFM to fuse the downscaled 250-m resolution MODIS LSTs at $t_b$ and $t_p$ and the 100-m resolution RTU LST product at $t_b$ to generate the 100-m resolution Landsat 8-like LST at $t_p$. In terms of the operating step, this method is similar to the three-step method. However, the three-step method adopted the optimal LST predictors to downscale MODIS LSTs at $t_b$ and $t_p$ to 250-m resolution, and then used the FSDAF to fuse the 100-m resolution RTU LST product at $t_b$ to generate the 100-m resolution Landsat 8-like LST at $t_p$. Concerning the STARFM-based fusion method, it used the STARFM to blend the RTU LST product at $t_b$ and the re-sampled 100-m resolution MODIS LSTs at $t_b$ and $t_p$ for generating the 100-m resolution Landsat 8-like LST at $t_p$. Furthermore, the RF-based LST downscaling used the RF regression model to downscale MODIS LST from 1-km resolution to 100-m resolution using the selected five predictors derived from the Landsat 8 PV image and GDEM data. Before all methods were implemented, the RTU LST product at $t_b$ was converted to the corresponding MODIS LST equivalent by establishing a simple linear transformation relationship between the MODIS LST product and RTU LST product at a 1-km resolution. This is because the LST images derived from Landsat 8 TIRS and Terra/MODIS differ obviously as a result of the instantaneous time difference in the local solar time and the sensor configuration of wavelength, signal-to-noise ratio, and viewing angles.

### 3.4. Accuracy Assessment

Ideally, to evaluate the performance of the predicted high-resolution LST, in-situ LSTs or actual LST images with the same resolution should be available as a reference. However, in practical cases, obtaining sufficient measured in-situ LSTs is very limited. In view of the higher accuracy of the RTU LST product, we used the RTU LST product at $t_p$ as the actual LST to evaluate the newly proposed framework. The root-mean-square error (RMSE), relative RMSE (RRMSE), and correlation coefficient (CC) were used as three evaluation indicators [46]. The RMSE and RRMSE can be used to evaluate the consistency between the predicted LST ($LST_{\text{pre}}$) and the true RTU LST product ($LST_{\text{true}}$), and CC can be used to characterize the spatial similarity degree between the predicted LST ($LST_{\text{pre}}$) and the actual RTU LST product ($LST_{\text{true}}$). In ideal circumstances, the closer the RMSE is to 0, the closer the predicted value is to the actual value; RRMSE well below 0.5 denotes that the used method is more accurate and reliable; the closer the CC is to 1, and the predicted image texture details are more similar to the actual image details. These three indices are depicted as follows:

$$\text{RMSE} = \left[\frac{1}{n}\sum_{i=1}^{n}\left(LST_{\text{pre}} - LST_{\text{true}}\right)^2\right]^{1/2} \tag{17}$$

$$\text{RRMSE} = \frac{\text{RMSE}}{\frac{1}{n}\sum\limits_{i=1}^{n} LST_{\text{true}}} \tag{18}$$

$$CC = 1 - \frac{\sum\limits_{i=1}^{n} \left(LST_{\text{pre}} - \overline{LST_{\text{pre}}}\right)^2}{\sum\limits_{i=1}^{n} \left(LST_{\text{true}} - \overline{LST_{\text{true}}}\right)^2} \tag{19}$$

where $n$ is the number of pixels in the LST image; $LST_{\text{pre}}$ is the predicted LST; $LST_{\text{true}}$ is the true RTU LST product; $\overline{LST_{\text{pre}}}$ is the mean of the predicted LST image; and $\overline{LST_{\text{true}}}$ is the mean of the RTU LST product.

## 4. Results

### 4.1. Selection Analysis of LST Predictors

The optimal selection of LST predictors plays an essential role in performing the spatial downscaling of MODIS LSTs. Taking Image A1 as a case, Table 2 lists the variate importance of fifteen predictors in the RF regression model. The %IncMSE denotes the percentage increase in the mean squared error (MSE), and the IncNode Purity denotes the increase in the tree node purity [18]. For convenient comparison, we also put forward the integrated contribution index (IC) by calculating the average value between %IncMSE and IncNode Purity to represent the synthetic importance of each parameter. It is very evident from Table 2 that, as two important parameters closely correlated with the LST distribution, PV and NDVI showed high ICs, and their values were 14,997.52 and 11,714.01, respectively. This is because the natural surfaces covered by large areas of vegetation play an important role in the regulation of LST by absorbing latent and sensible heat. The elevation impact on LST was also apparent with an IC of 5412.84 since the elevation presents a negative relationship with the LST in terrain with many mountainous landscapes [47]. Meanwhile, the soil dryness degree and soil moisture content impacts on the LST were also crucial, which can be represented through the bare soil index (BSI) and the soil humidity index (NDMI, SAVI). However, we found that variate importance was stronger in the aspect and LULC in terms of the MSE, with values of 1.58 and 2.04, respectively. It seems clear that the aspect in mountainous areas presented a pronounced impact on the solar illumination, and the land-cover type had a higher weighting of heat distribution in various areas.

**Table 2.** Variable importance of each factor in the RF regression for Image A1 (%IncMSE denotes the percentage increase in the mean squared error (MSE), IncNode Purity represents the increase in the tree node purity, and IC is integrated contribution index by calculating the average value between %IncMSE and IncNode Purity).

| Factors | %IncMSE | IncNode Purity | IC | Factors | %IncMSE | IncNode Purity | IC |
|---|---|---|---|---|---|---|---|
| PV | 14.855 | 29,980.19 | 14,997.52 | NDBI | 12.62 | 1942.49 | 977.55 |
| NDVI | 12.48 | 23,415.53 | 11,714.01 | Slope (°) | 23.13 | 1735.58 | 879.36 |
| Elevation (m) | 47.51 | 10,778.17 | 5412.84 | LSE | 18.47 | 1225.81 | 622.14 |
| NDMI | 10.21 | 8176.55 | 4093.38 | IEI | 6.01 | 1183.08 | 594.55 |
| BSI | 5.42 | 7164.63 | 3585.02 | MNDWI | 15.33 | 854.29 | 434.81 |
| SAVI | 5.82 | 4341.67 | 2173.75 | aspect | 1.58 | 561.02 | 281.30 |
| Longitude (°) | 44.10 | 3932.90 | 1988.50 | LULC | 2.04 | 45.36 | 23.70 |
| Latitude (°) | 38.30 | 3148.78 | 1593.54 | | | | |

Although the involved predictors have various importance in Table 2, the final selection of LST predictors mainly depends on the interaction of included factors. In the RF regression, removing or replacing some factors involved may change the importance scores because different inter-correlated variables could act as surrogates. Thus, in addition to referring to the %IncMSE, IncNode Purity, and IC values, the LST predictors need to be determined according to the regression fitting goodness of the RF model. Via incessant experiments, this paper selected the PV, elevation, slope, longitude, and latitude as LST predictors in the end. Similar to previous studies [6,39], the selected predictors had good representativeness in performing the LST downscaling and were shown to be key factors affecting LST. This result means that the vegetation biomass has a particularity during

the LST downscaling. Meanwhile, the spatial configuration of the terrain and geographic location (i.e., elevation, slope, longitude, and latitude) also played obvious roles in the LST downscaling since they determined the incident solar radiation that was available to heat the surface and the area's exposition to long-wave surface cooling [34].

However, for further discussing the reliability of the selection of LST predictors, Figure 4a,b also present the impacts of PV, elevation, slope, longitude, and latitude on the LST downscaling model for Image A1. With the RF algorithm, after the regression relationship between MODIS LST and selected predictors at a 1-km resolution was established, via assuming that the standard deviations (STD) of five predictors vary from 0.01 to 0.05, the white noise error with an average value of 0 and an STD of 0.01 to 0.05 was added to each factor. Then, we input the processed five LST predictors into the established RF regression relationship and estimated the LST in sequence. Later on, the LST estimated by corresponding predictors was evaluated using $R^2$ and RMSE by using the original MODIS LST as actual LST. To improve the modeling accuracy, after the outliers were removed, a total of 2000 pixel points were used to build the RF regression model.

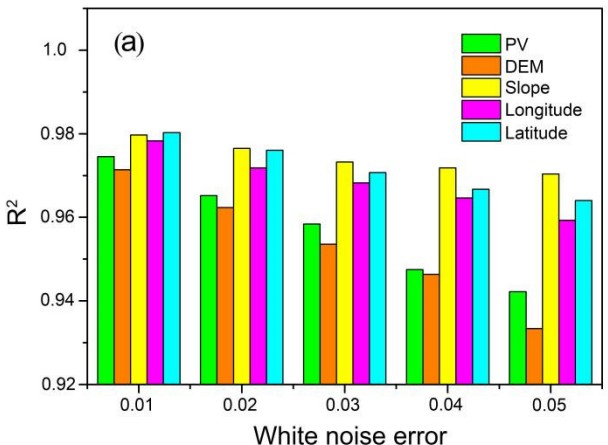 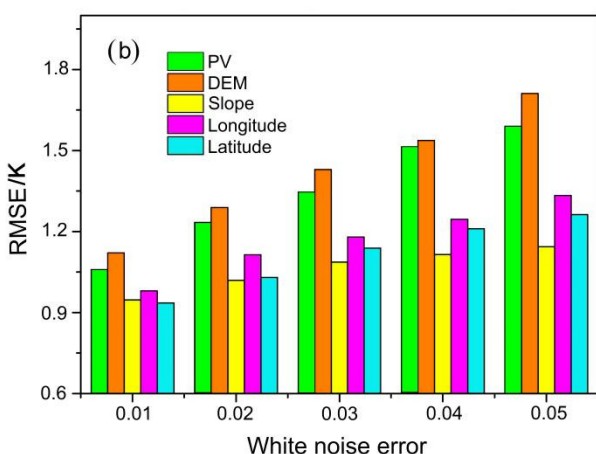

**Figure 4.** The $R^2$ and RMSE change in PV, elevation, slope, longitude, and latitude when five predictors have white noise errors from 0.01 to 0.05: (**a**) $R^2$, (**b**) RMSE. $R^2$ and RMSE values were calculated using the predicted LSTs and MODIS LSTs in Image A1.

As shown in Figure 4, the $R^2$ of five LST predictors presented an overall descending trend from 0.01 to 0.05 for white noise, and the change values were 0.038, 0.032, 0.019, 0.016, and 0.009 for elevation, PV, longitude, latitude, and slope, respectively. Their RMSEs displayed an ascending trend from 0.01 to 0.05 for white noise, and the highest RMSE difference (0.19 K) was found in the elevation. This finding indicates that the RF-based LST downscaling will generate a pronounced fluctuation if any errors exist in PV and elevation. Then, a poor estimation will be found in the subsequent fusion of the LST image. However, it is encouraging for the established RF model that the fitting goodness was acceptable at different noise situations ($R^2 > 0.93$); additionally, the final LST prediction accuracy was also within the rational range (RMSE < 1.72 K).

### 4.2. Accuracy Evaluation of the Framework

Figures 5 and 6 display the visual comparisons of the four 100-m resolution LST images predicted by the three-step method and three referenced methods with the actual RTU LST product for Images A1 and B1. Comparing the LST images, all methods can provide the LST with high spatial and temporal resolutions for the two study areas, but the specific details are different. The LSTs predicted by the three-step method (Figures 5b and 6b) more closely resembled the actual RTU LST product, presenting a relatively clear distribution pattern (e.g., the zoomed-in sub-regions). This indicates that blending the high spatial reconstruction from the regression and the daily temporal reconveyance from the image fusion process to predict high spatial and temporal resolution LSTs has obvious advantages.

Since the STARFM has some limitations in obtaining pure temporal information from the homogeneous pixels, the RF strategy (Figures 5c and 6c) and the STARFM-based fusion method (Figures 5d and 6d) showed unsatisfactory smoothing effects in the desert of Image A1 and the urban area of Image B1. However, the RF strategy was better than the STARFM fusion in the farmland of Image A1 and the forest of Image B1. In contrast, the RF-based LST downscaling (Figures 5e and 6e) exhibited the worst performance, showing many fragmented patches in the desert and cropland of Image A1. At the same time, when the land-cover type transitioned from one feature to another feature, obvious blocky artifacts were observed in both areas. Two possible reasons can explain this phenomenon. First, the RF-based regression process is based on the minimum mean square error. A common manifestation is that high values tend to be underestimated, and low values tend to be overestimated [6]. In addition, this phenomenon is caused by the LST variability in the coarse spatial resolution images, whereas the LST downscaling process of the RF algorithm does not fully consider this problem. In short, the three-step method showed an excellent visual agreement with the actual Landsat 8 LST, which also can be found in the frequency distribution maps of the four predicted LST images.

Figure 7a–c further displays the RMSE, RRMSE, and CC values between the actual RTU LST product and the LSTs predicted by the three-step method, RF strategy, STARFM-based fusion, and RF-based downscaling. Low RMSE and high CC values are indicative of LST prediction of satisfactory quality; the optimal method would result in the RMSE equal to 0 and CC value equal to 1. Taking study area A as a case, the mean RMSE decreased from 4.45 K for the RF-based downscaling to 1.89 K for the three-step method, the mean RRMSE decreased from 0.63 K for the RF-based downscaling to 0.28 K for the three-step method, and the mean CC increased from 0.828 for the RF-based downscaling to 0.976 for the three-step method. It is apparent that the RF-based LST downscaling method had the worst performance in study area A, the STARFM-based fusion method was better than the RF-based downscaling, and the three-step method and the RF strategy were better among the four methods. Especially for the three-step method, in Image A1, it had the best performance for three evaluation indicators: RMSE, RRMSE, and CC, of 1.62 K, 0.18 K, and 0.987, respectively. However, for study area B, the mean RMSE decreased from 3.02 K of the STARFM-based fusion to 1.30 K of the three-step method, the mean RRMSE decreased from 1.00 K of the STARFM-based fusion to 0.45 K of the three-step method, and the mean CC increased from 0.516 for the STARFM-based fusion method to 0.921 for the three-step method. It is worth noting that, in study area B, the RF-based downscaling presented higher accuracies than the RF strategy and STARFM fusion method, whereas its accuracy was still lower than the three-step method. One possible interpretation is that the STARFM performs poorly in keeping the texture of the LST image in heterogeneous regions (e.g., the urban area of Beijing) since the STARFM not fully considered land-cover change information. In contrast, the RF-based LST downscaling better retained the auxiliary information of LST predictors by constructing and averaging a large of randomized and de-correlated decision trees. Thus, the MODIS LST downscaling based on the RF regression has been widely used in previous studies.

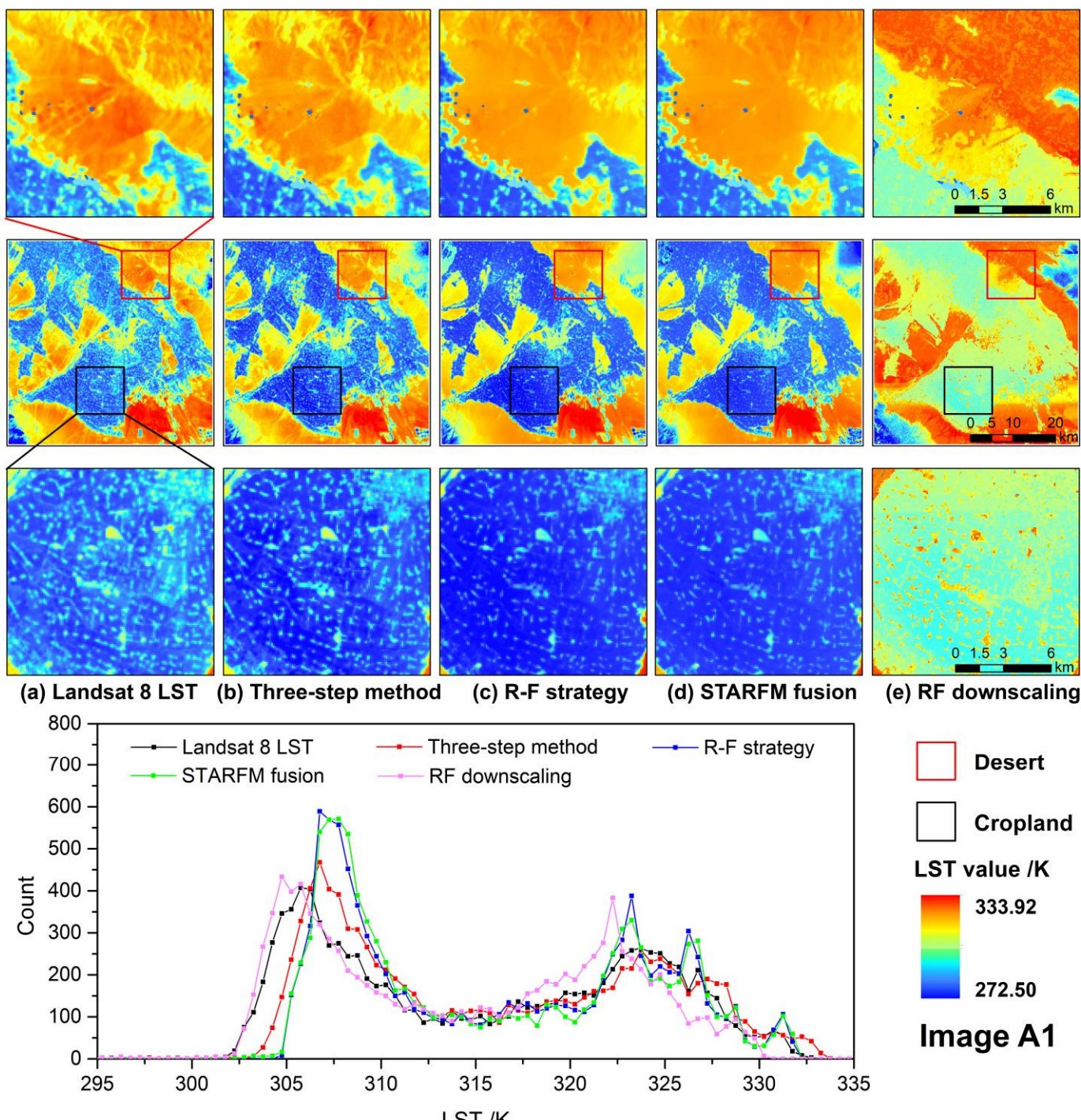

**Figure 5.** The spatial patterns and frequency distribution curves of predicted LSTs for Image A1: (**a**) Landsat 8 LST product; (**b**) the LST predicted by the three-step method, (**c**) the LST predicted by the RF strategy, (**d**) the LST predicted by the STARFM fusion, (**e**) the LST predicted by the RF downscaling.

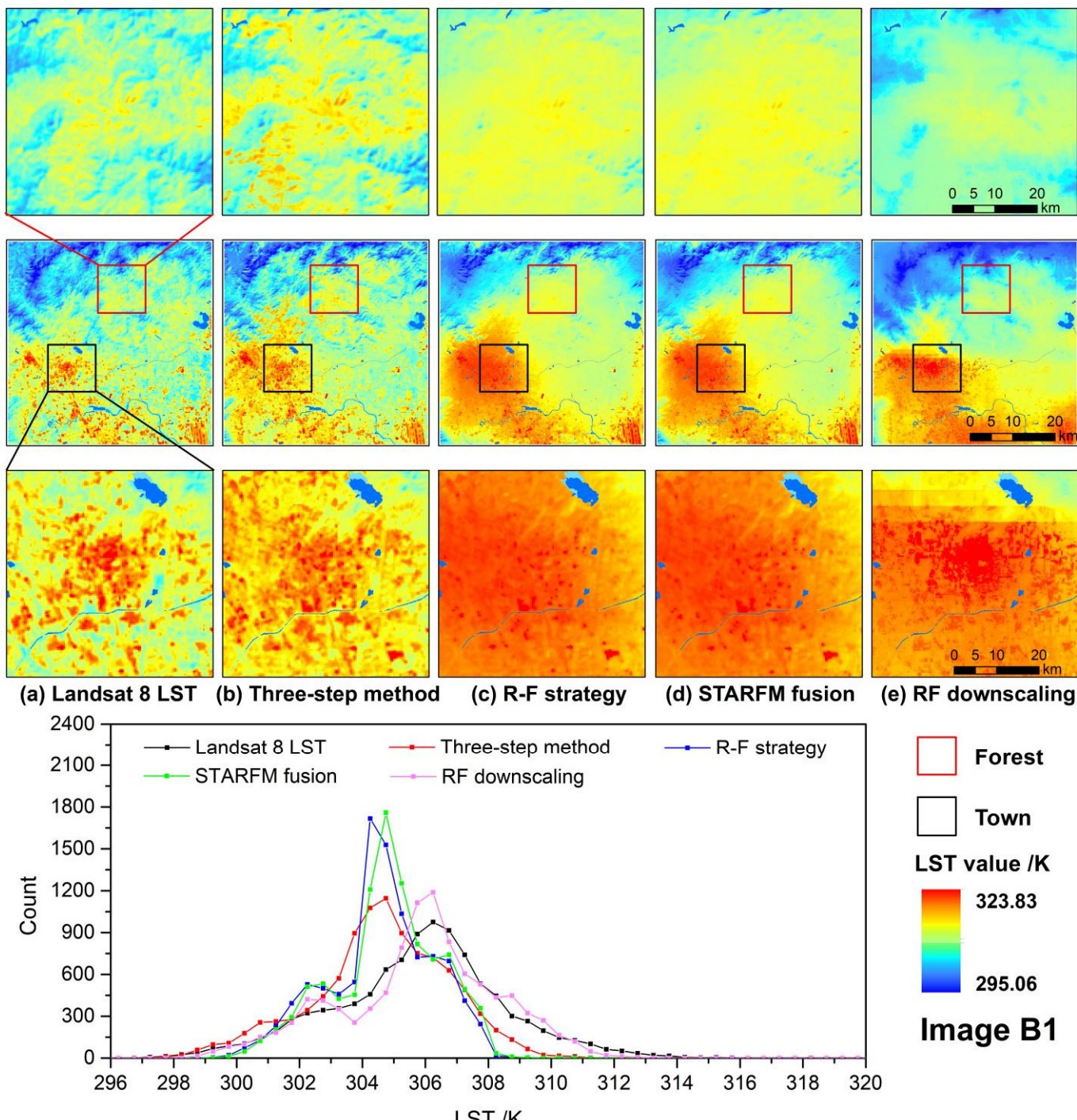

**Figure 6.** The spatial patterns and frequency distribution curves of predicted LSTs for Image B1: (**a**) Landsat 8 LST product; (**b**) the LST predicted by the three-step method, (**c**) the LST predicted by the RF strategy, (**d**) the LST predicted by the STARFM fusion, (**e**) the LST predicted by the RF downscaling.

From the above visual comparison and statistical analysis of different methods, an apparent superiority of the three-step method to other methods in some instances was observed. This is because the three-step method blended the information of multiple predicting factors in the downscaling of MODIS LST and also better considered the LST heterogeneity in the spatiotemporal fusion of LST.

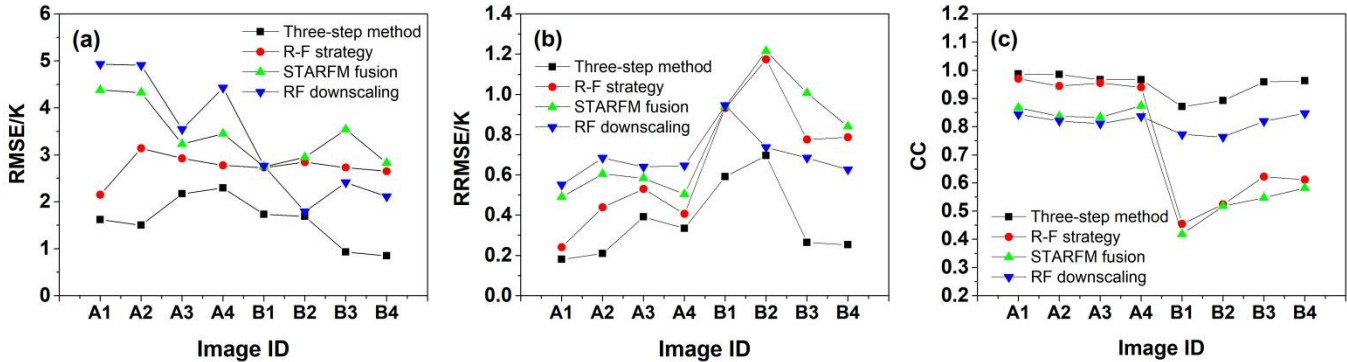

**Figure 7.** Accuracy comparisons of four kinds of LST prediction methods: (**a**) RMSE, (**b**) RRMSE, and (**c**) CC.

### 4.3. Distribution Error Analysis of Predicted LSTs

Taking the predicted LST images in Images A1 and B1 as cases, Figures 8 and 9 also present the distribution error maps of four predicted LSTs. The distribution error was defined as the absolute value of the spatial difference between the RTU LST product at $t_p$ and the predicted LST at $t_p$. This indicator can not only better reveal the distribution status of LST prediction error in the form of images but also reflect the actual size of the LST prediction error. Prior to the analysis, the distribution error of predicted LST was classified into five levels: 0~1 K, 1~2 K, 2~3 K, 3~5 K, and >5 K.

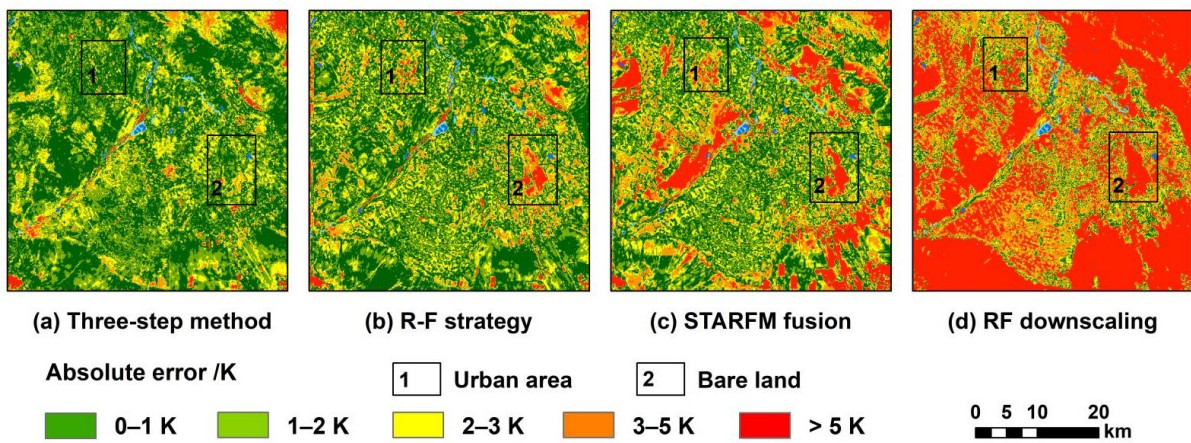

**Figure 8.** Distribution error maps of four predicted LST images for Image A1: (**a**) the three-step method, (**b**) the RF strategy, (**c**) the STARFM fusion, and (**d**) the RF downscaling.

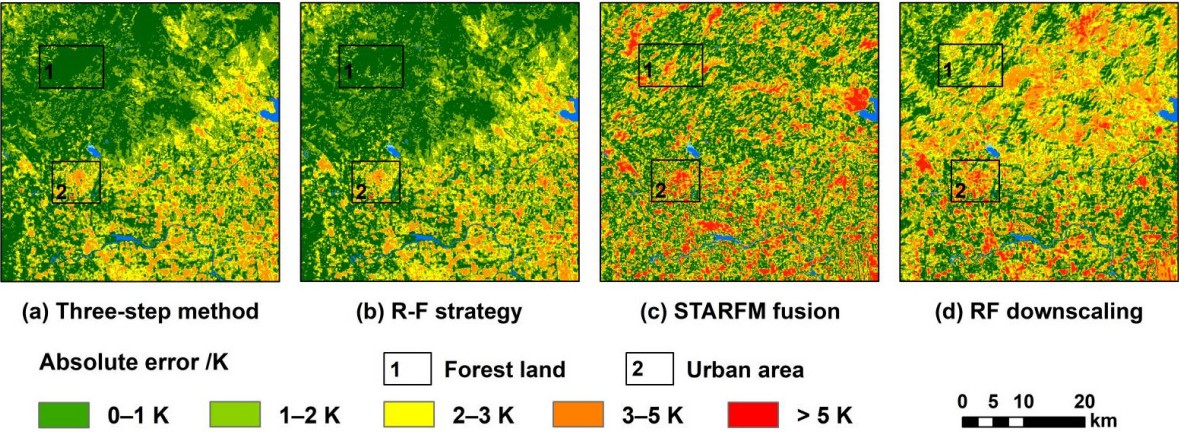

**Figure 9.** Distribution error maps of four predicted LST images for Image B1: (**a**) the three-step method, (**b**) the RF strategy, (**c**) the STARFM fusion, and (**d**) the RF downscaling.

Concerning Image A1, the distribution error of the three-step method was very similar to that of the RF strategy but observably different in level 4 and level 5. The land-cover types corresponding to these two kinds of levels were widely covered by impervious surfaces and sand lands, and they have peculiar thermal conductivity and heat capacity (see rectangle 1 and 2 of Image A1). The STARFM fusion method showed more errors in level 5, whereas it performed better than the RF-based LST downscaling. Regarding Image B1, the three-step method occupied the widest error area in level 1, followed by the RF strategy, STARFM-based fusion, and RF-based LST downscaling. The regions corresponding to level 1 were mainly distributed in mountainous areas, which were covered by trees (see rectangle 1 of Image B1). In addition, we found that the three-step method occupied the least error area in level 5, which was mainly located in the urban area (see rectangle 2 of Image B1).

For the sake of comparison, Table 3 shows in-detail statistics including the area percentages of these error levels for each method in Images A1 and B1. In Image A1, more than 43.96% of the study area fell under 1 K for the three-step method and generating the smallest percentage area in level 5, which was approximately 2.18% of this area. The RF strategy displayed a similar performance to the three-step method in level 1, with an error percentage of more than 39% of the area, whereas it had a larger error level (>3 K) than the three-step method, approximately 13.43% of study area A. In contrast, the STARFM fusion method performed poorly than the first two methods, with fewer area percentages in levels 1 and 2. The RF-based LST downscaling generated the most unsatisfactory result in level 5, displaying the largest error area percentage, approximately 55% of the study area. Similarly, as for Image B1, the three-step method still performs best, followed by the RF strategy, STARFM-based fusion method, and RF-based LST downscaling. For the three-step method, more than 42.30% of study area B fell below 1 K, and a minimum percentage area of 0.30% is generated outside of 5 K.

**Table 3.** Area percentage of distribution error for the four predicted LST images at five error levels in Images A1 and B1 (%).

| Image ID | Error Levels (K) | Three-Step Method | RF Strategy | STARFM-Based Fusion | RF-Based Downscaling |
|---|---|---|---|---|---|
| A1 | 0–1 | 43.96 | 39.06 | 26.86 | 7.04 |
| | 1–2 | 35.12 | 30.66 | 25.79 | 7.78 |
| | 2–3 | 14.24 | 16.80 | 17.26 | 9.05 |
| | 3–5 | 4.50 | 9.50 | 15.05 | 21.24 |
| | >5 | 2.18 | 3.93 | 15.02 | 54.89 |
| B1 | 0–1 | 42.30 | 43.26 | 30.05 | 21.83 |
| | 1–2 | 31.65 | 30.92 | 25.50 | 25.88 |
| | 2–3 | 16.50 | 15.76 | 18.54 | 24.18 |
| | 3–5 | 9.25 | 9.59 | 18.16 | 23.11 |
| | >5 | 0.30 | 0.47 | 7.75 | 5.00 |

However, due to the surficial property differences, such as the thermal inertia, pyroconductivity, and vegetation evaporation status, the performance of the proposed three-step method varies over land-cover types [48]. Thus, Tables 4 and 5 also discuss the mean distribution errors of predicted LSTs for Images A1 and B1 in five kinds of land-cover backgrounds. The comparisons in Images A1 and B1 show that all predicted LSTs performed poorly in the shrub and bare lands with many higher mean errors, whereas they performed better in the vegetation-covered regions (i.e., cultivated land, forest, and grassland). This indicates that all methods were suitable for the agricultural lands and forests but were not applicable to bare lands, especially for the STARFM-based fusion and RF-based LST downscaling. However, the three-step method still possessed good accuracy than the other methods in all land-cover types, and the RF-based LST downscaling was unsatisfactory in these regions. For instance, in Image A1, the three-step method generated the lowest mean distribution error value (1.84 K) in five land-cover types, and the RF strategy was second, with a mean distribution error of 2.56 K. The rest two kinds of methods were inapparent,

but the STARFM-based fusion presented better (with a mean distribution error of 3.31 K) than the RF-based LST downscaling (with a mean distribution error of 6.27 K).

**Table 4.** Average distribution errors of the four predicted LST images at five land-cover types in Image A1 (K).

| LULC Types | Three-Step Method | RF Strategy | STARFM-Based Fusion | RF-Based Downscaling | Mean Error |
|---|---|---|---|---|---|
| Cultivated land | 1.36 | 1.63 | 2.48 | 4.36 | 2.45 |
| Grassland | 1.59 | 2.02 | 3.91 | 9.14 | 4.16 |
| Shrub land | 1.62 | 1.67 | 2.07 | 3.40 | 2.19 |
| Artificial surface | 1.33 | 1.44 | 2.30 | 4.13 | 2.30 |
| Bare land | 3.30 | 6.07 | 5.78 | 10.3 | 6.36 |
| Mean error | 1.84 | 2.56 | 3.31 | 6.27 | |

**Table 5.** Average distribution errors of the four predicted LST images at five land-cover types in Image B1 (K).

| LULC Types | Three-Step Method | RF Strategy | STARFM-Based Fusion | RF-Based Downscaling | Mean Error |
|---|---|---|---|---|---|
| Cultivated land | 1.24 | 1.24 | 1.78 | 1.58 | 1.46 |
| Grassland | 1.00 | 0.98 | 1.83 | 2.03 | 1.46 |
| Shrub land | 1.07 | 1.08 | 1.81 | 2.66 | 1.65 |
| Artificial surface | 2.29 | 2.34 | 3.07 | 3.11 | 2.70 |
| Bare land | 1.38 | 1.62 | 1.64 | 2.77 | 2.43 |
| Mean error | 1.39 | 1.45 | 2.03 | 2.43 | |

## 5. Discussion

### 5.1. Impacts of MODIS LST Downscaling

In our study, the proposed three-step method combined the advantages of the regression-based LST downscaling and the FSDAF-based image fusion to generate LSTs with high spatiotemporal resolutions. Thus, the LST predicted by this new method is largely affected by the downscaling of MODIS LSTs at $t_b$ and $t_p$. Via implementing the forward fusion and the backward fusion with the data collected in two days in the same year, Figure 10a–d display the impacts of the re-sampled MODIS LSTs and downscaled MODIS LSTs on the predicted LSTs for all images. One input date pair is rewarded as the base time data at $t_b$, while the other is used as the prediction time data at $t_p$. The impact of the re-sampled MODIS LSTs on the predicted LST at $t_p$ can be denoted as the FSDAF-based fusion. The impact of the downscaled MODIS LSTs on the predicted LST at $t_p$ can be represented with the three-step method. Specifically, taking Figure 10a as a case, the forward fusion means to predict the LST on 21 July 2013 using the one LST data pair on 5 July 2013 and one MODIS LST on 21 July 2013. The backward fusion means to predict the LST on 5 July 2013 using the one LST data pair on 21 July 2013 and one MODIS LST on 5 July 2013.

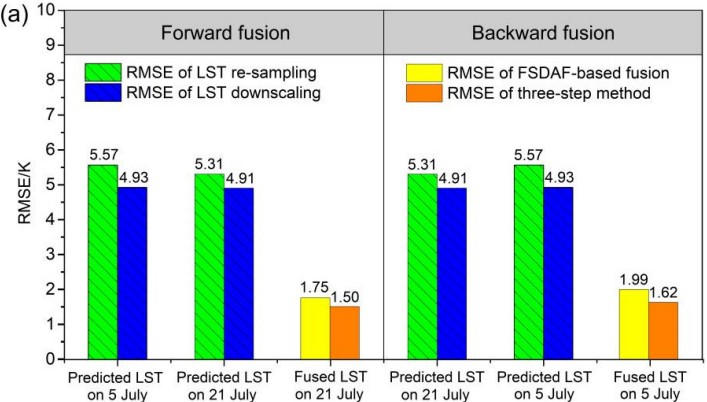

**Figure 10.** *Cont.*

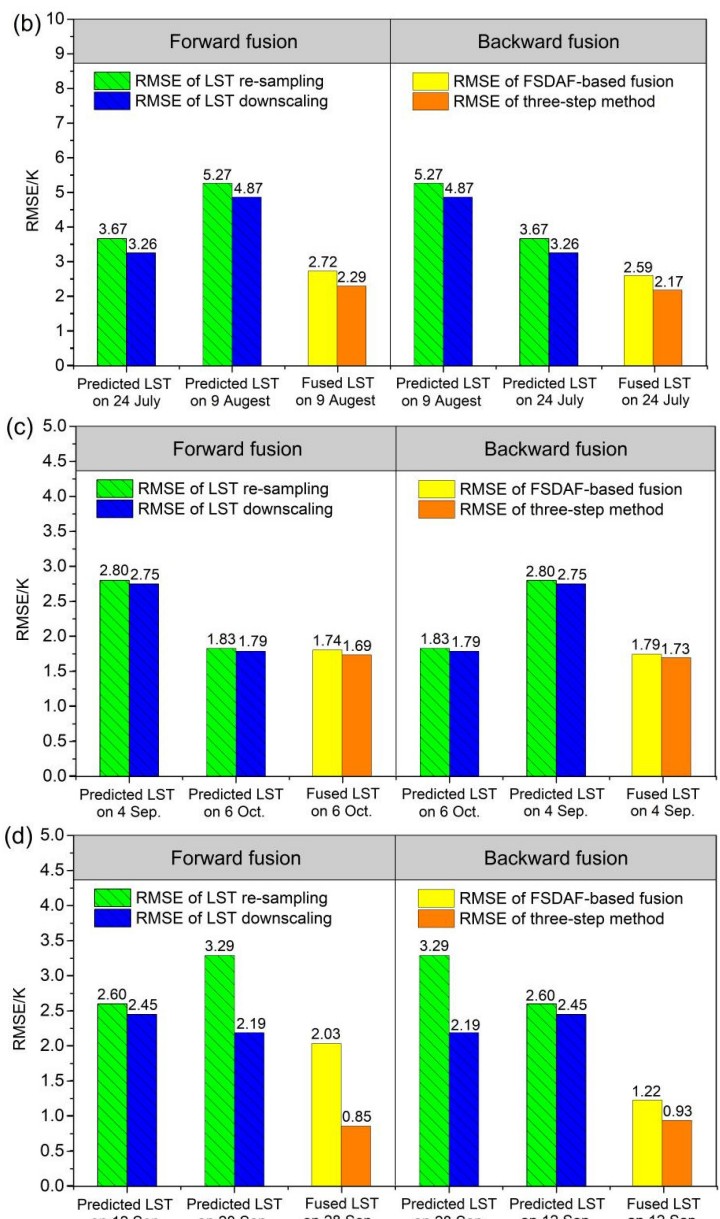

**Figure 10.** Impacts of two kinds of downscaling methods on the predicted LST image at $t_p$ during the forward fusion and the backward fusion in two days in the same year: (**a**) Images A1 and A2; (**b**) Images A3 and A4; (**c**) Images B1 and B2; (**b**) Images B3 and B4.

It is very evident from Figure 10a–d that no matter what fusion processes were used to predict the 100-m resolution LST at $t_p$, the three-step method that combines the downscaling of MODIS LST performed the best all the time. This is since the three-step method used the 250-m resolution auxiliary MODIS image as an intermediate resolution to sharpen the MODIS LST from 1 km to 250 m, and then to a 100-m resolution so that the downscaled MODIS LSTs at $t_b$ and $t_p$ was more accurate than the result of direct re-sampling, maintaining a wealth of LST change information from $t_b$ to $t_p$. Thus, it can be seen that, when using the downscaled MODIS LSTs at $t_b$ and $t_p$ to further fuse the 100-m resolution LST at $t_p$, the performance of the new framework is more pronounced in contrast with the FSDAF-based fusion. Accordingly, the downscaling accuracy of the MODIS LST lays a critical foundation for the final prediction of the 100-m resolution LST at $t_p$; the higher its accuracy is, the better the predicted LST image is. In addition, from Figure 10, a prominent finding indicates that, regardless of which methods were applied in predicting

the 100-m resolution LST at $t_p$, the more accurate the predicted LST at $t_p$ is, the better the fused LST at $t_p$ is, especially for the newly proposed method. We found from Figure 10a that LST RMSEs obviously decreased from 1.62 K of the backward fusion to 1.50 K of the forward fusion by using Image A1 and Image A2. Regarding the FSDAF-based fusion method, LST RMSEs decreased from 1.99 K of the backward fusion to 1.75 K of the forward fusion. This finding suggests that the downscaling accuracy of the MODIS LST at $t_p$ is more important and plays a crucial role in predicting the 100-m resolution LST at $t_p$.

*5.2. Advantages and Disadvantages of the Proposed Framework*

Similar to the traditional RF strategy, the proposed method is also an RF strategy, and it blended the spatial downscaling process of MODIS LST and the spatiotemporal data fusion process of Landsat 8 LST. However, the traditional RF strategy produced the 30-m resolution LST at $t_p$ by downscaling the Landsat LST (~100 m) into high-resolution (~30 m), and then used the STARFM to fuse the MODIS LST time series and the downscaled LST obtained [13,28]. The newly developed framework produced the 100-m resolution LST at $t_p$ by downscaling the MODIS LST (~1000 m) into medium resolution (~250 m) and then using the FSDAF to fuse the low-resolution LST time series and the downscaled LST obtained. The main difference between the two methods is the downscaling of MODIS LST and consideration of LST heterogeneity. In addition, for highlighting the importance of downscaling MODIS LST, the new method does not downscale the Landsat LST from 100-m to 30-m resolutions; thus, the three-step method only derived the 100-m resolution Landsat 8-like LST.

On the whole, compared with the previous RF strategies, the three-step method has more evident advantages for predicting high-resolution LST images in regions with strong spatial heterogeneity. First, the three-step method selected the optimal LST predictors by using the importance ranking to perform the MODIS LST downscaling so that reducing the multicollinearity between one and the other variables and improving the velocity of the model building since the redundant variables will substantially increase the complexity and computational cost of the model. Second, to capture more detailed change information of MODIS LSTs from the basic time $t_b$ to the prediction time $t_p$, by introducing the 250-m resolution LST predictors to downscale the MODIS LST, the three-step method effectively maintained the accuracy of MODIS LST and inherited the texture information of predictors. Third, by using the FSDAF algorithm, the three-step method can acquire LSTs with more clear textures in heterogeneous landscapes and predict 100-m resolution LST time series using daily 1-km resolution MODIS LST products and 100-m resolution Landsat 8 LST data.

Despite these advantages, the three-step method has several limitations. First, for better capturing the change information of MODIS LSTs from $t_b$ to $t_p$, the spatial downscaling of MODIS LSTs needs to take twice at $t_b$ and $t_p$. This process will take more time and cause some regression errors. Second, the selection of regression methods plays an essential role in the MODIS LST downscaling, which determines the texture and prediction accuracy of LST. Although the new method used the RF regression to build the non-linear relationship between LST and its predictors, more regression models still need to be considered in the future because the RF regression is limited by the number of samples to a great extent [47]. Third, the proposed three-step method only allows for the clear-sky condition because the thermal infrared signal cannot penetrate through clouds [49–51]. If we want to obtain the all-weather LST, it should be essential to remove the cloud effect. Considering that the combination of regression and data fusion has a number of potential applications in generating fine-resolution LST time series, we will propose some more accurate strategies for predicting the high spatiotemporal resolution LSTs in the future. A variety of spatial and temporal fusion models or algorithms could be adopted to enhance the texture characteristic of LST images, and more effective regression methods or machine learning algorithms could be used to improve the accuracy of LST downscaling.

## 6. Conclusions

By considering the spatial downscaling of MODIS LST and spatial heterogeneity of LST, this study developed a new framework (i.e., the three-step method) to predict the 100-m resolution Landsat 8-like LST at $t_p$ in two areas. Three key points are involved in this study: (1) the optimal selection of LST predictors; (2) the downscaling of MODIS LST; and (3) the implementation of the FSDAF algorithm. These processes can better solve the problems of inaccurate LSTs and unclear image textures and gain more detailed LST distribution features and more accurate LST values than other methods.

The visual comparison of the predicted LSTs derived from four kinds of methods indicates that the three-step method performed better than the other methods over heterogeneous regions, especially for the regions with relatively high LST variation and spatially fragmented landscapes, which obviously removed the blocky effect and blurring effect. With three evaluative indexes, our results presented similar results to the visual comparison, and the three-step method had the best accuracy: its RMSEs varied from 0.85 K of Image B4 to 2.29 K of Image A4, and RRMSEs varied from 0.18 K of Image A1 to 0.69 K of Image B2. Additionally, the distribution error analysis indicated that the three-step method minimized the predicted LST errors at five levels and five kinds of land-cover types, especially at bare land, with the minimum average distribution error (3.30 K of Image A1 and 1.38 K of Image B1, respectively). However, behind the use of the three-step method, there are still some limitations, such as the uncertainty of the LST downscaling model and the impacts of MODIS LST downscaling on LST prediction. As a result, to develop some more accurate LST prediction methods, more statistical regression models, spatiotemporal data fusion algorithms, and study areas need to be considered in the future.

**Author Contributions:** Conceptualization, X.Z. and X.S.; methodology, X.Z.; software, P.L.; validation, X.L.; data analysis, L.G.; writing—original draft preparation, X.Z.; writing—review and editing, D.G.; visualization, S.C.; funding acquisition, X.S. All authors have read and agreed to the published version of the manuscript.

**Funding:** This research was funded by the National Natural Science Foundation of China under Grant No. 41871242 and No. 42041005, the Second Tibetan Plateau Scientific Expedition and Research (STEP) program under Grant No. 2019QZKK0304, and the Fundamental Research Funds for the Central Universities.

**Institutional Review Board Statement:** Not applicable.

**Informed Consent Statement:** Not applicable.

**Data Availability Statement:** Not applicable.

**Acknowledgments:** The authors would like to acknowledge the National Aeronautics and Space Administration for the provision of MODIS product and ASTER GDEM data, and the United States Geological Survey for the provision of Landsat 8 remote sensing data, and the DATABANK Remote Sensing Data Engine for the provision of RTU LST product.

**Conflicts of Interest:** The authors declare no conflict of interest.

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
