# Peer review of "A Framework for Generating High Spatiotemporal Resolution Land Surface Temperature in Heterogeneous Areas"

_remotesensing, doi:10.3390/rs13193885_

Round 1

Reviewer 1 Report

The authors present a study to derive high spatiotemporal resolutions LSTs in heterogeneous areas, applied to two study areas in Zhangye and Beijing, China. The aim is to predict 100-m resolution Landsat 8-like LST images, starting from coarse MODIS LST maps. Specifically, the downscaling of MODIS LST and the spatiotemporal fusion of Landsat 8 LST is performed. The manuscript needs improvements, as detailed below:

-Why the authors chose the middle resolution as 250 m in the downscaling of MODIS LST? Please, specify.

-Section 3.3: it is not clear the difference between the proposed method and the R-F strategy. Please, specify.

-Table II: the symbols used must be explained in the caption. When applicable, the unit of measurements must be indicated in the table.

-Figure 4: the unit of measurements of RMSE is missing. The caption and the text must better describe how the R2 and the RMSE is computed (how many images, what is the truth, and so on)

-Figure 7: CC and SSIM provide the same information content, as evident from the behavior of the curves. So, reporting both metrics is not necessary.

-Lines 328-336. This part is not clear, hindering the repeatability for a reader. If I correctly understood, the prediction of fine-resolution Landsat-like PV at tp is necessary but it is not a simple issue, since the presence of vegetation can depend on unpredictable human factors and activities. How is the PV prediction performed?

-Table 4, 5: why a generic “LST distribution error” is reported, and not the RMSE as previously computed? The unit of measurement (K) is placed in a wrong place (it seems that LUCC are in K)

- The RMSE for a total image (or all the processed images) should be compared with the corresponding standard deviation (std) of the LST of the true/actual Landsat image. That is, the relative RMSE (ratio of the RMSE/std) must be computed to assess the reliability of the proposed method. For instance, ratios well below 0.5 are ascribed to accurate and reliable methods. In such a way, I can see if an error of 1 or 3 K is a good result or not: from the reported results, I can see only a comparison between methods, but not the accuracy level of the proposed approach.

-Are the results (red/yellow RMSE) of figure 10 also found for the other images/dates?

-Overall, the novelty of the proposed work is not clearly highlighted: what is the novelty with respect previous hybrid methods? It must be specified in the Introduction and the Discussion.

-Reference 37 and 38 are the same. Please, check the references.

Author Response

The authors are grateful to the reviewers for your careful and helpful comments. According to your main comments, many modifications have been performed in our manuscript, and we have also revised our manuscript format, tables, figures, and the use of the words, etc., carefully. Modifications have been highlighted using red text in the Revised Manuscript. Our responses to the comments are as follows, and the revised sections see the Revised Manuscript. 

1.Why the authors chose the middle resolution as 250 m in the downscaling of MODIS LST? Please, specify.

Ans: Thanks for your comments. Since previous hybrid methods usually re-sample the MODIS LST from 1-km resolution to the resolution within 100-m for the subsequent spatiotemporal fusion, this process will lose detailed spatiotemporal change information of LST. Fortunately, Terra/MODIS sensor can provide two kinds of resolutions PV images (i.e., 250-m and 500-m) every day to assist the downscaling of MODIS LST. Meanwhile, these PV images have the same instantaneous observation time as MODIS LST images. In our study, for better capturing the spatial texture information of LST change and enhancing the performance of LSTs fusion procedures in the following process, we chose the 250-m resolution PV image derived from Terra/MODIS to sharpen MODIS LSTs from 1-km to 250-m resolutions. This is because that the introduction of a 250-m resolution meets the requirement that the resolution difference is less than 3~5 times in the spatially non-uniform surfaces and offers more abundant spatiotemporal change information of LST in the process of LST downscaling. We have made a further explanation in Line 301-313.

2.Section 3.3: it is not clear the difference between the proposed method and the R-F strategy. Please, specify.

Ans: Thanks for your comments. Different from the previous R-F strategy, in our study, the R-F strategy used the RF regression model to downscale the MODIS LSTs at tb and tp to 250-m based on the NDVI, then adopted the STARFM to fuse the downscaled 250-m resolution MODIS LSTs at tb and tp and the 100-m resolution RTU LST product at tb to generate the 100-m resolution Landsat 8-like LST at tp. In terms of the operating step, this method is similar to the proposed method. But, the proposed method adopted the optimal LST predictors to downscale MODIS LSTs at tb and tp to 250-m resolution and then used the FSDAF to fuse the 100-m resolution RTU LST product at tb to generate the 100-m resolution Landsat 8-like LST at tp. The main difference between the two methods is the downscaling of MODIS LST and consideration of LST heterogeneity. In addition, for highlighting the importance of downscaling of MODIS LST, the new method does not downscale the Landsat LST from 100-m to 30-m resolutions; thus, the proposed method only derived the 100-m resolution Landsat 8-like LST. We have made a further explanation in Line 406-414 and Line 681-693.

3.Table II: the symbols used must be explained in the caption. When applicable, the unit of measurements must be indicated in the table.

Ans: Thanks for your comments. We have added the explanation in Line 481-483.

4.Figure 4: the unit of measurements of RMSE is missing. The caption and the text must better describe how the R2 and the RMSE are computed (how many images, what the truth is, and so on).

Ans: Thanks for your comments. We have revised figure 4. In addition, we also described the calculation of R2 and RMSE in detail in Line 484-495 and Line 506-508.

5.Figure 7: CC and SSIM provide the same information content, as evident from the behavior of the curves. So, reporting both metrics is not necessary.

Ans: Thanks for your comments. We have revised figure 7. In the revised manuscript, The root-mean-square error (RMSE), relative RMSE (RRMSE), and correlation coefficient (CC) were used as three evaluation indicators.

6.Lines 328-336. This part is not clear, hindering the repeatability for a reader. If I correctly understood, the prediction of fine-resolution Landsat-like PV at tp is necessary, but it is not a simple issue since the presence of vegetation can depend on unpredictable human factors and activities. How is the PV prediction performed?

Ans: Thanks for your comments. After the MODIS LST products at tb and tp were downscaled to 250-m resolution, the FSDAF algorithm was used to predict the Landsat 8-like LST product at tp in combination with the RTU LST product at tb. This part mainly describes the implementation process of LST fusion and does not predict the fine-resolution Landsat-like PV at tp.

7.Table 4, 5: why a generic “LST distribution error” is reported and not the RMSE as previously computed? The unit of measurement (K) is placed in the wrong place (it seems that LUCC is in K).

Ans: Thanks for your comments. In this paper, the LST distribution error is defined as the absolute value of the spatial difference between the RTU LST product at tp and the predicted LST at tp. This indicator can not only better reveal the distribution status of LST prediction error in the form of images, but also reflect the actual size of LST prediction error. Thus we mainly used this indicator in Tables 4 and 5 to discuss the statistic values of LST prediction errors at various land-cover types. The unit of measurement (K) has been revised in Line 637-638. 

8.The RMSE for a total image (or all the processed images) should be compared with the corresponding standard deviation (std) of the LST of the true/actual Landsat image. That is, the relative RMSE (ratio of the RMSE/std) must be computed to assess the reliability of the proposed method. For instance, ratios well below 0.5 are ascribed to accurate and reliable methods. In such a way, I can see if an error of 1 or 3 K is a good result or not: from the reported results, I can see only a comparison between methods, but not the accuracy level of the proposed approach.

Ans: Thanks for your comments. We have added the relative RMSE (ratio of the RMSE/std) to assess the reliability of the proposed method. We have made a further explanation in Line 551-552, Line 560-562, and figure 7.

9.Are the results (red/yellow RMSE) of figure 10 also found for the other images/dates?

Ans: Thanks for your comments. The results (red/yellow RMSE) of figure 10 are suitable for the other images/dates. In figure 10, we have added more data to explain this result. More explanation is described in Line 644-655.

10.Overall, the novelty of the proposed work is not clearly highlighted: what is the novelty with respect to previous hybrid methods? It must be specified in the Introduction and the Discussion.

Ans: Thanks for your comments. We have specified the novelty of the proposed work in the Introduction and Discussion section. More explanation is described in Line 133-138 and Line 694-707.

11.Reference 37 and 38 are the same. Please, check the references

Ans: Thanks for your comments. We have revised and checked the references again.

Reviewer 2 Report

The proposed work present improvement in the methodology to downscale Land Surface Temperature at a target time from low resolution sensors (such MODIS) with high revisit time to high resolution (100 m) LST (Landsat 8-like). To do so, they need a pair of synchronous high (L8) and low (MODIS) images and a set of products associated to MODIS data such (vegetation indices, land cover class, DEM etc..) at low and medium resolution to learn predictors for downscaling.

The paper is well written. However, the large number of acronyms can make following the method and understanding it tedious at times. To improve the fluidity of reading, I would avoid the acronyms CLST and HLST which have little meaning since the method is mainly based on MODIS (LST at 1km) and all its products from the visible (NDVI, NDWI etc..). And on Landsat which is the only sensor to propose data synchronous to MODIS, LST and high resolution.

The only comments I will have are:

- what is the difference between RF and R-F?

L231 : could you detail FSDAF algorithm and develop acronym ?

Figure 2 : where is used the result of step1 : predictors at 1km ? Is it in step 2 ? an arrow is missing…

The proposed work is of interest to the LST data user community. I think this work can be published with minor revisions.

Author Response

The authors are grateful to the reviewers for your careful and helpful comments. According to your main comments, many modifications have been performed in our manuscript, and we have also revised our manuscript format, tables, figures, and the use of the words, etc., carefully. Modifications have been highlighted using red text in the Revised Manuscript. Our responses to the comments are as follows, and the revised sections see the Revised Manuscript. 

1.The paper is well written. However, the large number of acronyms can make following the method and understanding it tedious at times. To improve the fluidity of reading, I would avoid the acronyms CLST and HLST which have little meaning since the method is mainly based on MODIS (LST at 1km) and all its products from the visible (NDVI, NDWI etc..). And on Landsat which is the only sensor to propose data synchronous to MODIS, LST and high resolution.

Ans: Thanks for your comments. We have revised these acronyms in the revised manuscript. 

2.what is the difference between RF and R-F?

Ans: Thanks for your comments. The RF (random forest) is the RF-based regression downscaling, and the R-F is the R-F (regression-then-fusion) strategy. The RF-based method used the RF regression model to downscale MODIS LST from 1-km resolution to 100-m resolution using the selected five predictors derived from the Landsat 8 PV image and GDEM data. The R-F strategy used the RF regression model to downscale the MODIS LSTs at tb and tp to 250-m based on the NDVI, then adopted the STARFM to fuse the downscaled 250-m resolution MODIS LSTs at tb and tp and the 100-m resolution RTU LST product at tb to generate the 100-m resolution Landsat 8-like LST at tp. More explanation is described in Line 406-425.

3.L231 : could you detail FSDAF algorithm and develop acronym?

Ans: Thanks for your comments. The FSDAF algorithm denotes the flexible spatiotemporal data fusion algorithm. More explanation is described in Step 3: Spatiotemporal image fusion of LST.

4.Figure 2: where is used the result of step1: predictors at 1km? Is it in step 2? an arrow is missing…

Ans: Thanks for your comments. We have revised figure 2.

Round 2

Reviewer 1 Report

The authors improved the manuscript, following my comments and adding the further metric RRMSE. It is suitable for publication, after addressing the following minor comments:

-Table 2: When applicable, the unit of measurements must be indicated in the table. For instance, what is the unit of measurements of elevation, slope, latitude, longitude

-Table 4, 5: The unit of measurement (K) near LULC should be removed.

Author Response

-Table 2: When applicable, the unit of measurements must be indicated in the table. For instance, what is the unit of measurements of elevation, slope, latitude, longitude?

-Table 4, 5: The unit of measurement (K) near LULC should be removed.

And: Thanks for your comments. We have revised Tables 2, 4, 5.
